# Clinical and Pathological Profiles of Vertebral Bone Metastases from Endometrial Cancers: Evidence from a Twenty-Year Case Series

**DOI:** 10.3390/diagnostics12122941

**Published:** 2022-11-25

**Authors:** Stefano Bandiera, Francesca Salamanna, Veronica Borsari, Deyanira Contartese, Marco Bontempi, Matilde Tschon, Giovanni Tosini, Stefano Pasini, Silvia Terzi, Milena Fini, Alessandro Gasbarrini

**Affiliations:** 1Spine Surgery Unit, IRCCS Istituto Ortopedico Rizzoli, 40136 Bologna, Italy; 2Complex Structure of Surgical Sciences and Technologies, IRCCS Istituto Ortopedico Rizzoli, 40136 Bologna, Italy; 3Scientific Direction, IRCCS Istituto Ortopedico Rizzoli, 40136 Bologna, Italy

**Keywords:** bone metastasis, vertebral bone, endometrial cancer, clinical-pathological characteristics, survival

## Abstract

Patients with endometrial cancer (EC) frequently have metastases to lungs, extra-pelvic nodes, and liver. Although an uncommon occurrence, cases of EC metastasis to bone, prevalently in vertebral bone, have also been reported. The objective of this study was to analyze clinical and pathological profiles of patients with EC metastatic to vertebral bone. We carried out a retrospective case series on surgically treated patients for this pathology. From 2001 to 2021, out of 775 patients with bone metastasis, 1.6% had bone metastasis from EC. The median time between the diagnosis of primary tumor and that of bone metastases was 31.5 months. Solitary bone lesion was present in 7 patients and lumbar vertebrae were the segments most affected. Pathological fractures in 46.2% of patients and spinal pain in all were present. In terms of location, 46.2% of bone metastases resided within the anterior section of the vertebra, while the remaining presented an extension within the anterior and posterior sections, with 46.1% of cases showing an extradural extra-osseous extension and paraspinous envelope. Median survival after diagnosis of bone metastasis was 11.5 months. Vertebral bone metastasis in EC is a rare phenomenon, with severe prognosis. An in-depth understanding of this topic may guide future management and treatment decisions, thus improving life expectancy and quality.

## 1. Introduction

Endometrial cancer (EC) is the fourth most common cancer in women, with 604,000 new cases reported worldwide in 2020 and a mortality of 7.7% [1,2]. By examination of age, it is diagnosed beginning at the age of 20 to more than 85 years with a peak at age 40–50 [3]. At diagnosis, about 13% of women with EC cancer are at an advanced stage, which results in reduced patient survival [4,5]. However, the improvements in EC management techniques including advanced imaging modalities, novel surgical techniques, personalized radiotherapy, and chemotherapy have led to an increase in the survival of patients. In turn, the prolonged survival has led to a corresponding increase of patients with metastases and recurrence. The most frequent sites of distant metastasis in EC are the lungs, extra-pelvic nodes, and liver. EC bone metastases are uncommon, and the rate is estimated to be <1% [6]. The most common site for EC bone metastasis is the vertebral column, followed by the pelvis and long bones [7,8]. The causes of this ”preferentially spread” have not yet been fully understood, but it has been hypothesized that the cellular and molecular characteristics of cancer cells and the tissues to which they metastasize are critical and affect the pattern of metastatic spread. Furthermore, it was also hypothesized that this pattern of metastatic spread to bone reflects the distribution of the so-called red bone marrow, a highly vascular tissue containing hematopoietic stem cells and an active microenvironment that promotes cellular growth [9]. Once cancer spreads to the bone, it is associated with a wide range of morbidities including pain, increased fracture risk, and hypercalcemia, thus seriously affecting quality of life, performance status, and independent functioning, as well as survival. Yoon et al. found that in patients with EC bone metastasis, advanced stage and initial multiple bone metastases had significantly shorter free survival [10]. Several studies reported a median survival time of 7–12 months among women with EC and bone metastases, and more than 60% of these patients died within 6 months after being diagnosed with bone metastases [11,12,13]. Although treatments of bone metastases are rarely curative, the disease control is often possible using systemic anticancer treatments on a background of multidisciplinary supportive care. This care involves the use of bone-targeted agents able to inhibit tumor-associated osteolysis and prevent skeletal morbidity as well as the use of suitable local treatments such as radiation therapy and specialist palliative care to reduce the effect of metastatic bone disease on physical functioning. However, impending or existing pathological fractures (i.e., skeletal-related events—SRE), spinal cord compression, unbearable pain, and resistance to radiotherapy are indications/complications that in most cases require an orthopedic surgical approach [14]. For instance, the suggestion for surgery in EC vertebral bone metastases depends on symptoms and surgical approaches differ according to the site and size (segments and layers) of the metastasis. A palliative decompression associated to stabilization and radiotherapy upon wound healing is usually recommended [15]. However, in patients with a relatively favorable prognosis, more aggressive interventions including total en bloc resection can also be recommended and the pain related to vertebral body fractures can be treated via vertebroplasty or kyphoplasty [15]. Despite numerous therapeutic and surgical approaches, to date only a limited number of case-reports and information are available in literature for EC vertebral bone metastases. In this article, we describe the clinical features of our patients who were diagnosed with EC vertebral bone metastases, including vertebral bone metastasis location, Weinstein–Boriani–Biagini classification (segments and layers), symptoms, surgical management, treatment, complication, follow-ups, and clinical outcomes.

## 2. Materials and Methods

The present study is a retrospective case-series performed in accordance with relevant guidelines and regulations, performed from the data available from the Department of Oncological and Degenerative Spine Surgery of our Institution and approved by the Local Ethics Committee of the Emilia Romagna Region (Comitato Etico Indipendente Area Vasta Emilia Centro, CE-AVEC, Prot. n. 0007902, 7 January 2019). Informed consent was obtained from all participants. The study involved a total of 775 patients with vertebral bone metastases who were surgically treated and managed from 2001 to 2021. Out of them, 13 patients had EC vertebral bone metastasis. In these patients’ indications for surgery were (1) presence of progressively worsening pain, not controlled by painkillers; (2) presence of spinal instability; (3) progressive or abrupt neurologic deterioration due to spinal cord compression, in case of failure of conservative therapy (such as radiotherapy and chemotherapy); and (4) estimated survival of more than 3 months and a general condition that allowed surgery. All surgical procedures were performed by 3 senior surgeons.

An extensive review of medical records was performed for all patients to obtain specific clinical data and overall outcomes. Patient demographics, age, body mass index (BMI), primary EC histological type and grade, comorbidities, smoking habits, menopausal status, and pre-operative hematological parameters at diagnosis of bone metastasis were reviewed and collected. Furthermore, specific data on bone metastases, i.e., bone metastasis diagnostic tool, time from EC primary diagnosis to bone metastasis, main bone lesion level, presence of other bone metastasis lesions, presence of fracture, presence of extraosseous metastases, pre- and post-operative pain, pre- and post-operative chemo-radiotherapy, pre- and post-operative neurologic function, surgical staging of bone metastasis and months to death/last follow-up after bone metastasis and last follow-up status were reviewed and collected.

Pre- and post-operative pain was evaluated using a numeric rating scale (NRS): each patient was asked to grade the pain from 0 = no pain to 10 = highest pain experienced [16]. 

The pre- and post-operative and the latest follow-up neurologic functions were evaluated using the Frankel Grade, according to the degree of spinal cord injury, as follows: Grade A (complete loss of sensation and motor function below the level of injury); Grade B (no motor function, but some sensation is retained below the level of the lesion); Grade C (some muscles below the level of injury have the motor function, but no proper function present); Grade D (proper function present below the plane of injury, walking with crutches); Grade E (typical motor and sensory function, pathological reflexes possible) [17].

Surgical staging of bone metastasis was evaluated according to Weinstein–Boriani–Biagini classification [18]. In brief, the spine is radially divided into 12 equal radial segments (clock-face) in the axial plane and examined in 5 layers from the superficial to deep plane (Figure 1).

All data were collected in a Microsoft Excel spreadsheet. Statistical analyses were made using MATLAB(R) (R2018a, the MathWorks, Natick, MA, USA). Results were reported as mean ± standard deviation (SD) if normally distributed, or median (interquartile range (IQR)). A survival curve was estimated using the Kaplan–Meier estimator [19,20]. Furthermore, to provide some estimate of the probability of survival of the population, the curve was also interpolated using a Weibull distribution. The interpolation was performed using a maximum likelihood estimation algorithm [21].

## 3. Results

### 3.1. Patients’ Characteristics and Pathological Profiles

In the current study, 13 patients (1.6%) with EC vertebral bone metastases were included. Out of the thirteen patients, only two of them were younger than 50 years (age range: 31–75). The median age of the patients was 62 (IQR 56.0–69.0) (Table 1). BMI (kg/m^2^) data, present for six patients, ranged from 18 to 36, with one underweight patient, two overweight patients and one patient with class 1 obesity. Clear cell carcinoma was present in 30.7% of patients on histological examination, while the remaining had an adenocarcinoma. Most patients (76.9%) had histological Grade 3 for primary tumor at diagnosis, while the others had Grade 1 (15.4%) and 2 (7.7%). At the time of EC bone metastases, 46.1% of patients had one or more comorbidities, and hypertension was the most common (66.6%) in these patients. No comorbidities were present for 30.7% of patients and three patients’ data on comorbidities were not reported. Two patients were smokers at the time of EC bone metastases, seven patients had never smoked and four patients’ data on smoking habits were not reported. All patients were in menopausal status. At the time of EC bone metastases diagnosis, altered blood hematological parameters were detected, i.e., white blood cells (WBC), hematocrit, hemoglobin (Hb), and lymphocytes, as well as C-reactive protein (CRP) and γ-glutamyltransferase (GGT). Four patients’ data on hematological parameters were not reported.

### 3.2. Bone Metastases

For nine patients, the median time between the diagnosis of primary tumor to the diagnosis of bone metastases was 31.5 months (IQR 14.25–54), while for the remaining four patients, data were not reported (Table 2). Bone metastases were diagnosed via at least 2 investigative tools, including positron emission tomography (PET), bone biopsy, or computed tomography (CT), or in fewer cases with magnetic resonance imaging (MRI) and plain radiography (RX) (Table 2). The most common vertebral site was the lumbar one (76.9%) followed by thoracic (23.1%) sites (Table 2). Seven patients (53.8%) had solitary metastatic bone lesion, while the other six (46.2%) had multiple bone lesions (Table 2). Two patients also had extra-osseous metastases that were detected along with bone metastases diagnosis. Pathological fractures were present in 46.2% of patients and spinal pain (back pain, radiating pain, or both) was presented in all patients before spinal surgery (Table 2). The mean value of spinal pain before surgery, measured via NRS, was 4.2 ± 1.5, while after surgery almost all patients reported a major relief of pain with a NRS of 1.1 ± 1.1, showing a statistically significant pain reduction after surgery (*p* = 0.0006) (Table 2). In total, 46.2% of patients received pre-operative chemo- and/or radiotherapy, while a higher percentage of patients (61.5%) also received a post-operative chemo- and/or radiotherapy (Table 2). Pre-operative Frankel classification provides an assessment of spinal cord function and demonstrated that most patients (69.2%) had a Grade E (typical motor and sensory function, pathological reflexes possible), 15.4% had a Grade D (proper function present below the plane of injury, walking with crutches) and 7.8% had a Grade C (some muscles below the level of injury have the motor function, but no proper function) (Table 2). Concerning the post-operative Frankel classification, it was of Grade E in 69.2% patients and Grade D in 30.8% patients. At the last follow-up, all patients had a Frankel classification E, except for three patients because data were not present (Table 2). Concerning the Weinstein–Boriani–Biagini classification, the analysis of intervertebral extension found that 46.2% of bone metastases resided only within the anterior region of the vertebral element (sectors 4–9). The remaining patients (53.8%) presented a bone metastasis extension within the anterior and posterior area (within sectors 4–9, 1–3, and/or 10–12). No tumors were found that resided only within the posterior vertebral area. When these lesions were analyzed according to the tissue layers involved, it was found that 46.1% of cases demonstrated an extradural extra-osseous extension and paraspinous envelope (layers A, B, C, and D). In two cases (15.4%), there was an intradural extra-osseous extension into the spinal canal and paraspinous envelope (layers A, B, C, D, and E), while in two patients (15.4%), an extra-osseous extension only into the paraspinous region, i.e., layers A, B, and C, was detected. Finally, one patient had an extra-osseous extension only into the spinal canal, i.e., layers B, C, and D, one had an intraosseous deep (layer C) extension and extraosseous extradural (layer D), and one had only an extraosseous soft tissue extension (layer A) (Table 2). The mean follow-up period was 17.5 months. Local disease control was present in 38.4% of patients, 23.0% patients had a local recurrence, 30.7% of patients died, and one patient dropped out from follow-up. All deaths were due to systemic or local disease progression and none of them were associated with intraoperative mortality (Table 2). The median survival after diagnosis of bone metastasis was 11.5 months (range, 0–81 months).

### 3.3. Surgical Approaches and Variables

As previously reported, pain symptoms due to instability or fracture were present in all patients treated in our series and surgical treatment was performed specifically for progressive pain or when either vertebral collapse or metastatic growth caused spinal cord compression. In one case, the surgery was palliative, while in the remaining cases, intralesional curettage of “active” forms and extracapsular curettage (or en bloc resection) in “aggressive” forms was used (92.3%) (Table 3). The aim of these techniques was the same: restoration of spinal stability and decompression of neural structures, i.e., spinal cord and nerve roots. In our case series, the most-used approach was the posterior (84.6%) followed by the anterior and combined approaches (15.4%) (Table 3). Spacers, i.e., titanium cages, cement, allografts, were used in 53.8% of cases (Table 3). In 92.3% of patients, the plane of dissection had transgressed into the lesion (intralesional margin) with contamination of resected margins in all cases, while in one patient a wide surgical margin with a plane of resection within normal tissue was performed (Table 3). Two patients had complications. In one patient there was a dural injury that was detected intraoperatively and repaired surgically; a second patient had a late postoperative complication, i.e., vertebral fracture under the instrumentation (Table 3).

### 3.4. Patients’ Survival Estimates

Statistical results for the construction of the survival curve using the Kaplan–Meier method are shown in Table 4. The last column shows the survival probability of the population at the time of death of some patient. The estimated probabilities, on the other hand, represent the percentages of the population currently deceased and surviving at the time of death of some member. A plot of the curve is shown in Figure 2. The solid line represents the experimental survival curve formed from the results shown in Table 4. The dashed curve is the future projection provided by interpolation with a Weibull distribution. The characteristic parameters of the distribution as assessed using interpolation are shown in Table 5. Table 5 also shows that the coefficient of determination [22] is quite low (R^2^ = 0.45), indicating a weak statistical correlation between the survival curve and the estimate. This is due to the limited number of cases in our study and to the high survival of these patients. Looking at the data in Table 2, only one patient survived 81 months. The other survivors entered the cohort later and are still alive. Based on these considerations, it has been estimated that 50% of the population will survive up to 29 months of follow-up entry and 25% of patients up to 51 months.

## 4. Discussion

EC is one of the most common women’s cancers, with steadily increasing incidence that seems to be correlated with obesity, comorbidities, and fragility [23,24]. Bone metastases from EC are relatively rare, and because of this, reports present in the literature are described within the context of all bone metastases, regardless of their location. Despite the low reported incidence of bone metastasis from EC, autopsy series have shown a significantly higher incidence of about 35% [25,26], emphasizing the need of accurate clinical surveillance and screening after primary cancer. Furthermore, it was reported that the vertebral column represents the most common site of EC bone metastasis [7,8]. Despite these data, to date, only one case-series on six patients reported the clinicopathological characteristics, surgical outcomes, survival, and complications following surgery for vertebral bone metastases from different gynecological cancers, i.e., cervical cancer, EC, and leiomyosarcoma [27]. However, in this era of rapid progress and advanced technologies, it would be mandatory to better understand and analyze individually each of the sites of primary gynecological malignancy that cause vertebral bone metastasis to thus have a more “personalized” approach to patient management. The purpose of this study was to analyze the clinical and pathological profiles of patients with EC metastatic to vertebral bone.

In our series, we found EC vertebral bone metastasis in 1.6% of patients; patients with age ranging from 31 to 75 years, with a prevalence of Grade 3, were present in this series. As reported in recent studies, and differently from old studies, most of the patients of our series (53.8%) had solitary metastatic bone lesion, and this is probably due to the advances in diagnostic techniques that have led to an increase in earlier diagnosis [28]. Additionally, in accordance with previous reports, the median time between the diagnosis of EC to the diagnosis of bone metastases was 31.5 months [29,30]. Lumbar vertebrae are the most common involved vertebrae and the anterior region or the anterior associated to posterior one represent the specific locations for bone metastasis, with most patients presenting an extradural extra-osseous extension and paraspinous envelope. No patient presented metastasis in the posterior location alone. These data suggest that endometrial cancer tends to metastasize in specific segments of the vertebral column and in specific regions and levels within. Although some studies evaluated spinal metastases from gynecological cancer, no authors have investigated which specific vertebral areas and levels are affected by bone metastasis. The involvement of a specific vertebral region is a key datum not only for surgical planning, but also could also be useful for prognosis. 

Because vertebral bone metastases can lead to bone instability and spinal cord compression, these patients can present with intractable pain, impaired ambulatory ability, and neurologic dysfunction. More than 90% of patients with spinal metastases reported pain and approximately 20% had cord compression [31]. In our case series, the major clinical presenting symptom was pain (100%), with pathological fractures present in 46.2% of patients [32]. Since the surgical treatment of spinal metastatic tumors is important to relieve pain, restore neurologic function, and restore the immediate and permanent stability of the spine, in our series all patients were submitted to surgery [33]. While historically restricted to cases with acute neurological impairment, a landmark study from 2005 has shown unparalleled benefits of surgical treatment for the patient’s functional autonomy and mobility. In our patients’ series, surgery was safe and effective in improving pain and neurological symptoms; complications (15.4%) and local recurrence (23.0%) were present only in a small patient’s percentage. Pain relief and the low incidence of local recurrence may be also due to the post-operative chemotherapy and/or radiotherapy that were administered to 62% of patients. As extensively reported in the literature, while radiation therapy results in local pain relief, chemotherapy tackles the further spread of the disease [29]. Yoon et al. also reported that survival after bone metastasis was longer in the patients who received radiotherapy (with/without chemotherapy) than in the patients who received chemotherapy alone as a salvage therapy [11,12,28]. For our series, the survival function estimates that 50% of the population will survive up to 29 months of follow-up entry and 25% of patients up to 51 months. These data differ slightly from those present in the literature, where it was seen that once diagnosed with EC bone metastases, most patients died within 1 year [22]. However, it is important to underline that these studies examined length of survival of all patients with bone metastases rather than survival of those with spine metastases alone and considered different types of gynecological cancers all together. Furthermore, the variation in survival function as compared to the literature may be also due to differences with respect to grading and treatment of the primary tumor and vertebral bone metastasis, and baseline health of the patients at time of surgery, including comorbidities and blood tests.

We acknowledge that this study had some limitations. First, as a retrospective study, potential biases were unavoidable. Second, the number of cases was relatively small, but due the rarity of EC vertebral metastases, a large-sample study could not be performed. For this reason, our study did not include subgroup analyses of histological subtypes or other variables. A large-sample multi-center study would probably be necessary. Third, the number of accessible variables provided for some patients were limited. For example, some information on BMI, comorbidities and blood tests, type of adjuvant chemotherapy, details of chemotherapy drugs and sequence of treatments were missing. 

In conclusion, our study shows that, although rare, EC vertebral bone metastases continue to gain relevance in the wake of an ever-growing life expectancy of patients, underlining that increased attention should be paid to the mechanisms and prognostic value of site-specific EC bone metastases. The study, for the first time, examined a series of patients with a specific gynecological primary tumor, EC, that metastasized to a specific bone site, vertebrae. Our findings on bone metastasis localization and on the outcomes of surgical procedures can provide important information for the treatment of these patients, since the wide heterogeneity of primary cancer and of bone metastases require personalized and targeted treatments by type of primary tumor and site of metastasis. The results of this study may help clinicians form the metastatic workup of these cancers, relate important prognostic information to patients, and provide insight on the most likely gynecologic malignancy by metastatic site at presentation.

## Figures and Tables

**Figure 1 diagnostics-12-02941-f001:**
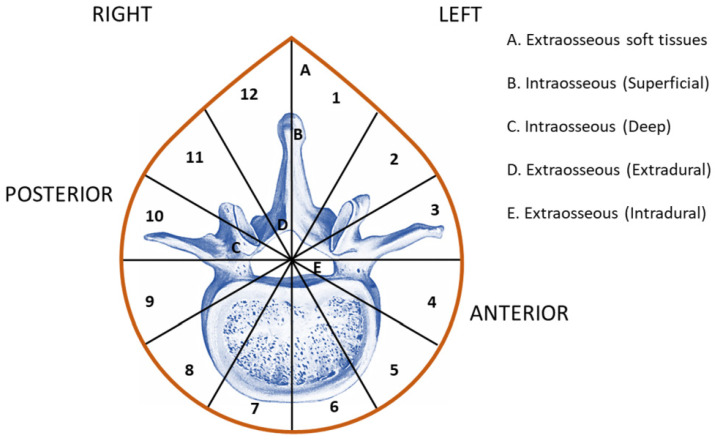
The Weinstein–Boriani–Biagini classification of spinal tumors. The location of the lesion is described using the 12 radiating regions (1–12) and 5 concentric layers (A–E), with layers D and E representing epidural and intradural involvement, respectively.

**Figure 2 diagnostics-12-02941-f002:**
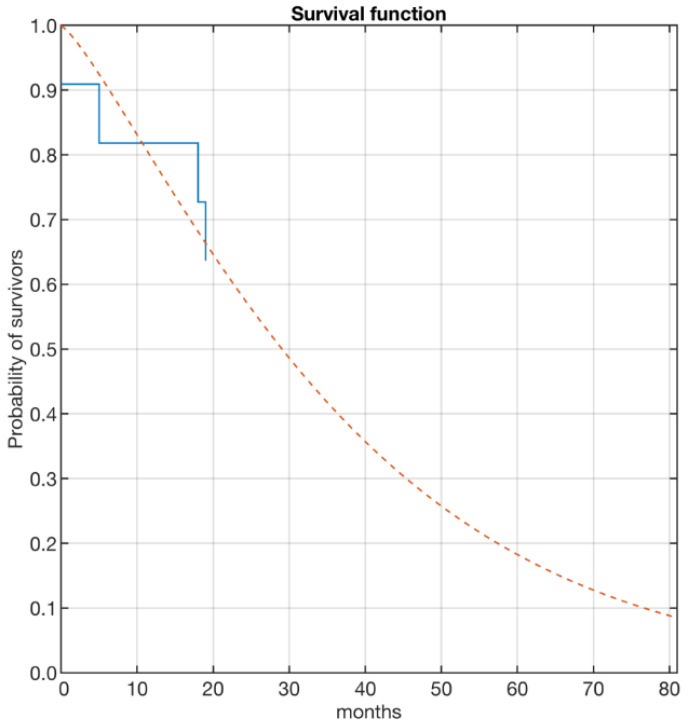
Plot of the survival curve of the treated patient population. Blue line = experimental curve; Orange line = interpolation.

**Table 1 diagnostics-12-02941-t001:** Patients’ characteristics and pathological profiles.

Patient	Age—Years	BMI	Primary Tumor Histological Cell Types	Primary Tumor Histological Grade	Comorbidity	Smoking	Menopausal Status	Pre-Operative Hematological Parameters
1	69	Not reported	Adenocarcinoma	G3	Not reported	Not reported	Yes	Not reported
2	66	Not reported	Adenocarcinoma	G3	Not reported	Not reported	Yes	Not reported
3	73	Not reported	Clear cell carcinoma	G3	Not reported	Not reported	Yes	Not reported
4	58	Not reported	Adenocarcinoma	G3	None	Not reported	Yes	Not reported
5	31	Not reported	Adenocarcinoma	G3	None	No	Yes (surgically induced)	↑ WBC; ↓ potassium, PLT, MPV
6	53	Not reported	Clear cell carcinoma	G3	None	No	Yes	↑ MCV, MCH; ↓ WBC
7	75	Not reported	Adenocarcinoma	G3	Hypertension, gonarthrosis	No	Yes	↑ LDH, WBC, Hb, fibrinogen
8	46	36.7	Clear cell carcinoma	G1	Hashimoto’s thyroiditis	No	Yes	↑ CRP, WBC, RDW, neutrophils; ↓ lymphocytes
9	67	22.9	Adenocarcinoma	G1	Facioscapulohumeral muscular dystrophy, diabetes, hypertension, hypercholesterolemia	No	Yes	↑ RDW, PLT, eosinophils, GGT; ↓ lymphocytes
10	62	25.7	Adenocarcinoma	G3	Hypertension	Yes	Yes	↑ ALP, GGT, CRP, neutrophils; ↓ RBCs, hematocrit, Hb, lymphocytes, PLT
11	59	28.5	Adenocarcinoma	G2	Hypertension, hypercholesterolemia	No	Yes	↓ RBCs, Hb, hematocrit, lymphocytes, CRP; ↑ eosinophils, ALP, GGT
12	73	20.9	Adenocarcinoma	G3	CVI, renal insufficiency, diverticulosis, chronic multifactorial anemia	No	Yes	↑ WBC, neutrophils, monocytes, CRP, GGT, fibrinogen; ↓ RBCs, Hb, hematocrit, lymphocytes
13	56	18	Clear cell carcinoma	G3	None	Yes	Yes	↓ RBCs, Hb, hematocrit, lymphocytes, eosinophils; ↑ neutrophils

Abbreviations: BMI: body mass index; WBCs: white blood cells; RBCs: red blood cells; PLT: platelet; MPV: mean platelet volume; RDW: red blood cell distribution width; MCV: mean corpuscular volume; MCH: mean corpuscular hemoglobin; Hb: hemoglobin; CRP: C-reactive protein; GGT: γ-glutamyltransferase, CVI: chronic venous insufficiency; ↑: increase; ↓: decrease.

**Table 2 diagnostics-12-02941-t002:** Clinicopathologic characteristics of bone metastasis from endometrial cancers.

Patient	Bone Metastases Diagnosis	Time from Primary Diagnosis to Bone Metastasis (Months)	Main Bone Lesion Level	Other Bone Metastasis Lesions	Presence of Fracture	Extraosseous Metastases	Pain	Pre- and Post-Operative NRS (0–10)	Pre-Op Chemo-Radiotherapy	Frankel Pre-Op	Frankel Post-Op	Weinstein–Boriani–Biagini Classification	Post-Op Chemo-Radiotherapy	Frankel at Latest FU	Months to Dead/Last FU after Bone Metastasis, Status
1	PET, CT, bone biopsy	Not reported	L4	No	No	Not reported	Yes	Not valuable	No	D2	D3	3–11 A–D	No	E	5, dead
2	PET, CT, bone biopsy	Not reported	L4	No	Yes	Not reported	Yes	4 1	No	E	E	2–8 A–E	Chemo-radiotherapy	E	19, dead
3	PET, CT, bone biopsy	Not reported	L3	Yes	No	No	Yes	Not valuable	No	E	E	4–6 C–D	Radiotherapy	E	26, alive with local disease control
4	PET, CT, bone biopsy	Not reported	L3	Yes	Yes	Not reported	Yes	5 1	Radio-chemotherapy	C	D2	2–11 A–D	Chemo-radiotherapy	Unknown	Unknown
5	RX, MRI, CT	12 months	T11	Yes	No	No	Yes	4 1	Radiotherapy	E	E	1–6 B–D	No	E	81, alive with local recurrence
6	PET, RX, CT	48 months	L2	Yes	Yes	No	Yes	4 0	No	E	E	3–6 A–D	Chemotherapy	E	4, alive with local disease control
7	RMN, CT and bone biopsy	Not reported	T6	No	Yes	No	Yes	3 1	No	Not reported	E	2–9 A–E	Chemo-radiotherapy	E	18, dead
8	PET, MRI, CT	72 months	L4	No	No	No	Yes	Not valuable	No	E	E	12–7 A–D	Chemotherapy	E	22, alive with local recurrence
9	Bone biopsy, RX, CT	96 months	L2	Yes	No	No	Yes	4 4	No	E	D2	4–5 A	Radiotherapy	Unknown	0, dead
10	Bone biopsy, CT	6 months	L1	Yes	No	Yes	Yes	2 1	Radio-chemotherapy	E	E	7–9 A–C	Radiotherapy	E	22, alive with local recurrence
11	Bone biopsy, CT	15 months	L5	No	Yes	No	Yes	4 1	Radiotherapy	E	E	4–9 A–C	No	E	1, alive with local disease control
12	MRI, CT	48 months	T12	No	No	Yes	Yes	4 0	Radio-chemotherapy	D3	D3	1–12 A–D	No	Unknown	5, alive with local disease control
13	PET, RX, CT	15 months	L5	No	Yes	No	Yes	8 1	Radio-chemotherapy	E	E	4–6 A–D	No	E	4, alive with local disease control

**Table 3 diagnostics-12-02941-t003:** Endometrial cancer bone metastasis surgical approaches.

Patient	Surgery Type	Surgical Approach	Spacer Use	Surgery Margins	Contaminated Margins	Surgical Complications
1	Palliative	Posterior	None	Intralesional	Yes	No
2	Curettage-intracapsular	Posterior	Titanium cage	Intralesional	Yes	No
3	Curettage	Posterior	Cement	Intralesional	Yes	No
4	Curettage	Posterior	Cement	Intralesional	Yes	No
5	Curettage-intracapsular	Posterior	Allograft	Intralesional	Yes	No
6	Curettage-intracapsular	Posterior	None	Intralesional	Yes	No
7	Curettage	Posterior	None	Intralesional	Yes	No
8	Curettage-intracapsular	Posterior	Cement	Intralesional	Yes	No
9	Curettage-intracapsular	Anterior	None	Intralesional	Yes	No
10	En bloc wide	Anterior + posterior	Custom made titanium cage	Wide	No	Late post-operative: lower flat depression of L3 and L4 vertebrae, degeneration under the instrumentation
11	Curettage	Posterior	None	Intralesional	Yes	No
12	Curettage	Posterior	None	Intralesional	Yes	Intra-operative: dural injury
13	Curettage	Posterior	Cement	Intralesional	Yes	No

**Table 4 diagnostics-12-02941-t004:** Summary table of patient survival data.

Time of Event (Month)	Number of Deaths	Live Patients	Estimated Probability		Survival Probability at Time of Event
			Death	Survival	
0	1	11	0.0909	0.9091	0.9091
5	1	10	0.1000	0.9000	0.8182
18	1	9	0.1111	0.8889	0.7273
19	1	8	0.1250	0.8750	0.6364

**Table 5 diagnostics-12-02941-t005:** Table of interpolation parameters used for survival estimation. The coefficient of determination and the parameters of the Weibull distribution are shown. The last two columns show an estimate of the first half-life (t_½_) and second half-life (t_¼_) of the population.

R^2^	Scale (λ)	Shape (k)	t_½_ (month)	t_¼_ (month)
0.45	39.0649	2.4746	29	51

## Data Availability

The datasets generated during and/or analyzed during the current study are available from the corresponding author on reasonable request.

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
