# Peer review of "Clinical and Pathological Profiles of Vertebral Bone Metastases from Endometrial Cancers: Evidence from a Twenty-Year Case Series"

_diagnostics, 2022, doi:10.3390/diagnostics12122941_

Round 1
Reviewer 1 Report
To learn on Vertebral Bone Metastases from Endometrial Cancers which is a rare condition, is without doubt, very important issue for gynecologists. By rule endometrial cancer is more likely to metastasize to the lymph nodes, liver and lungs. Bone metastasis spreads hematogenously and is rarely reported in the literature with an incidence in less than 1% of cases. Thus, the recognition of symptoms and presentation of bone metastases is of utmost importance.
Patients’ characteristics, as well as specific data on bone metastases are thoroughly analyzed. From orthopedic point of view this is a very useful article. Bone metastases diagnostic tool, time from EC primary diagnosis to bone metastasis, main bonelesion level, presence of other bone metastasis lesions, presence of fracture, presence of extraosseous metastases, Surgical approaches and variables such as pre- and post-operative pain, pre- and post-operative chemo-radiotherapy, pre- and post-operative neurologic function, surgical staging of bone metastasis and months to dead/last follow-up after bone metastasis and last follow-up status were reviewed and collected.
For a gynecologist the importance of this study is to keep in mind the possibility for spreading of endometrial cancer to vertebral bone and to pay the attention on all signs and symptoms during the follow-up.
However, as a gynecologist I have some questions and concerns related to the presented series of patients.
1. In the studied series of patients with vertebral bone metastases authors presented 13 cases of endometrial cancer, which is 1.6% from a total of 755 patients with vertebral bone metastases (both male and female). It would be, however, important to know which percentage of all endometrial cancers in the studied period
2. 30.7% of patients had a squamous cell carcinoma on histological examination, while the remaining had an adenocarcinoma. Squamous cell carcinoma is not the histological entity for endometrial cancer and indicates rather presence of cervical cancer, than of endometrial cancer. Also adenocarcinoma can be endometrial or endocervical by origin. This is not clarified. If primary origin of a squamous cancer is cervix, this would change a lot the results and discussion, since cervical cancer does have the potential to metastasize on bones.
3. The stage of endometrial cancer is not analyzed not the type of gynecological surgery. Almost all cancers are grade 3, which would require not only hysterectomy but also pelvic and para-aortal lymphadenectomy. Stage II of endometrial cancer would require radical hysterectomy. Omitting lymphadenectomy as a part of surgical treatment for endometrial cancer
The study is good for the first time, examining a series of patients with a specific gynecological primary tumor, EC, that metastasized to a specific bone site, vertebrae. For the purpose of scientific quality of the study parameters related to the site, type and stage of gynecological tumor should be provided.
If it could be of any help, here is the latest reference on this subject
Wang J, Dai Y, Ji T, Guo W, Wang Z, Wang J. Bone Metastases of Endometrial Carcinoma Treated by Surgery: A Report on 13 Patients and a Review of the Medical Literature. Int J Environ Res Public Health. 2022 Jun 2;19(11):6823. doi: 10.3390/ijerph19116823. PMID: 35682407; PMCID: PMC9180500.
Author Response
Comments and Suggestions for Authors
To learn on Vertebral Bone Metastases from Endometrial Cancers which is a rare condition, is without doubt, very important issue for gynecologists. By rule endometrial cancer is more likely to metastasize to the lymph nodes, liver and lungs. Bone metastasis spreads hematogenously and is rarely reported in the literature with an incidence in less than 1% of cases. Thus, the recognition of symptoms and presentation of bone metastases is of utmost importance.
Patients’ characteristics, as well as specific data on bone metastases are thoroughly analyzed. From orthopedic point of view this is a very useful article. Bone metastases diagnostic tool, time from EC primary diagnosis to bone metastasis, main bone lesion level, presence of other bone metastasis lesions, presence of fracture, presence of extraosseous metastases, Surgical approaches and variables such as pre- and post-operative pain, pre- and post-operative chemo-radiotherapy, pre- and post-operative neurologic function, surgical staging of bone metastasis and months to dead/last follow-up after bone metastasis and last follow-up status were reviewed and collected.
For a gynecologist the importance of this study is to keep in mind the possibility for spreading of endometrial cancer to vertebral bone and to pay the attention on all signs and symptoms during the follow-up.
However, as a gynecologist I have some questions and concerns related to the presented series of patients.
We thank the reviewer for the comments on the manuscript and we are very proud that there is a gynecologist to review the manuscript, we are sure that this can enrich our work to make it interesting not only for orthopedists but also for gynecologists.
In the studied series of patients with vertebral bone metastases authors presented 13 cases of endometrial cancer, which is 1.6% from a total of 755 patients with vertebral bone metastases (both male and female). It would be, however, important to know which percentage of all endometrial cancers in the studied period.
We agree with the reviewer that it would be important to know the percentage of all endometrial cancers in the period studied, however the study object of the manuscript was conducted in a mono-specialist (orthopedic) Hospital (IRCCS Istituto Ortopedico Rizzoli), and we are unable to give the percentage of primary tumors. However, literature data estimated that in 2015 about 160 new cases of endometrial cancer were diagnosed in our Region (Emilia-Romagna), i.e. 7.0 cases per 100,000 resident women, with an age-standardized rate of 5.1 per 100,000.
30.7% of patients had a squamous cell carcinoma on histological examination, while the remaining had an adenocarcinoma. Squamous cell carcinoma is not the histological entity for endometrial cancer and indicates rather presence of cervical cancer, than of endometrial cancer. Also adenocarcinoma can be endometrial or endocervical by origin. This is not clarified. If primary origin of a squamous cancer is cervix, this would change a lot the results and discussion, since cervical cancer does have the potential to metastasize on bones.
We thank the reviewer for the comment. We apologize very much but have mistakenly reported ‘squamous’ instead of ‘clear’. We have changed the text.
In addition, we recontrolled all the histological examination and we confirm that all carcinomas were of endometrial origin.
The stage of endometrial cancer is not analyzed not the type of gynecological surgery. Almost all cancers are grade 3, which would require not only hysterectomy but also pelvic and para-aortal lymphadenectomy. Stage II of endometrial cancer would require radical hysterectomy. Omitting lymphadenectomy as a part of surgical treatment for endometrial cancer
The stage of endometrial cancer and the type of gynecological surgery were not analyzed as data were not available for all patients but only for ⁓45% of them. However, as suggested by the reviewer, patients with a complete medical record and with a grade 3 tumors were all treated with hysterectomy and pelvic lymphadenectomy.
The study is good for the first time, examining a series of patients with a specific gynecological primary tumor, EC, that metastasized to a specific bone site, vertebrae. For the purpose of scientific quality of the study parameters related to the site, type and stage of gynecological tumor should be provided.
The available parameters related to the site, type and stage of gynecological tumor were reported in Table 1.
If it could be of any help, here is the latest reference on this subject
Wang J, Dai Y, Ji T, Guo W, Wang Z, Wang J. Bone Metastases of Endometrial Carcinoma Treated by Surgery: A Report on 13 Patients and a Review of the Medical Literature. Int J Environ Res Public Health. 2022 Jun 2;19(11):6823. doi: 10.3390/ijerph19116823. PMID: 35682407; PMCID: PMC9180500.
We thank the reviewer for the suggestion. We added the reference (now reference 24) in the manuscript discussion.
Reviewer 2 Report
I read with great interest the Manuscript titled "Clinical and pathological profiles of vertebral bone metastases from endometrial cancers: evidence from a twenty-year case series".The manuscript is interesting and the topic falls within the scope of the journal, however, some points may be improved.
Methodology is accurate and conclusions are supported by the data analysis.
The entire manuscript should be further revised by a native English speaker to improve clarity and readability and should be checked in order to correct typos (i.e. line 92: BMI).
Patient number 5 in table 1 is in menopause despite being 30 years of age. Is the data correct? What was the cause of menopause? I think the data is interesting and the concept needs to be clarified.
Discussions can be expanded and improved by citing relevant articles (I suggest authors to consider the following article PMID: 35793638; 31657456).
I think it could be of interest for the readers of Diagnostics, and, in my opinion, it deserves the priority to be published after minor revisions.
Author Response
Comments and Suggestions for Authors
I read with great interest the Manuscript titled "Clinical and pathological profiles of vertebral bone metastases from endometrial cancers: evidence from a twenty-year case series".
The manuscript is interesting and the topic falls within the scope of the journal, however, some points may be improved.
Methodology is accurate and conclusions are supported by the data analysis.
The entire manuscript should be further revised by a native English speaker to improve clarity and readability and should be checked in order to correct typos (i.e. line 92: BMI).
After the reviewer suggestion, we checked the manuscript in order to correct typos.
Patient number 5 in table 1 is in menopause despite being 30 years of age. Is the data correct? What was the cause of menopause? I think the data is interesting and the concept needs to be clarified.
Patient number 5 had a surgically induced menopause derived from the primary endometrial carcinoma. After the reviewer suggestion we added this data on table 1.
Discussions can be expanded and improved by citing relevant articles (I suggest authors to consider the following article PMID: 35793638; 31657456).
As suggested by the reviewer we added the references 23 and 24 to the discussion section.
I think it could be of interest for the readers of Diagnostics, and, in my opinion, it deserves the priority to be published after minor revisions.
Reviewer 3 Report
This is an interesting analysis concerning a rare problem of bone metastases in endometrial cancer. The authors have analyzed 775 patients treated because of endometrial cancer during 20 years (2001-2021), 13 of the patients had bone metastases. The authors present a detailed clinical and pathological analysis of the endometrial cancer patients, the type of the treatment and survival rate.
There is scarce number of publications related to this problem. Similiar work was published by Ucella et al, 2013, Gynecol Oncol and concerned also 13 primary bone metastatic endometrial cancers, totally 19 patients, the analysis was related to patients treated between 1984-2001, in last century.
This is a well written study, of high clinical impact.
I suggest a correction of Table 2, in the upper line of table the text fuses and is difficult to read.
Author Response
Comments and Suggestions for Authors
This is an interesting analysis concerning a rare problem of bone metastases in endometrial cancer. The authors have analyzed 775 patients treated because of endometrial cancer during 20 years (2001-2021), 13 of the patients had bone metastases. The authors present a detailed clinical and pathological analysis of the endometrial cancer patients, the type of the treatment and survival rate.
There is scarce number of publications related to this problem. Similiar work was published by Ucella et al, 2013, Gynecol Oncol and concerned also 13 primary bone metastatic endometrial cancers, totally 19 patients, the analysis was related to patients treated between 1984-2001, in last century.
This is a well written study, of high clinical impact.
I suggest a correction of Table 2, in the upper line of table the text fuses and is difficult to read.
We thank the reviewer for the comments and for the suggestion on Table 2. It is true that the first row of table 2 is difficult to follow. We have placed the table horizontally to make it more readable.
Round 2
Reviewer 1 Report
Dear Authors,
I appreciate your clear and honest replies. I understand that it is difficult to obtain all relevant data in a mono-specialist hospital, particularly when being far away from female genital pathology. The fact that you pointed out the risk and significance of vertebral bone metastases will be important for gynecologists.
Thank you for making the corrections where possible. I agree with revised version of your article.